# Reward learning from human preferences and demonstrations in Atari

**Borja Ibarz**
DeepMind
bibarz@google.com

**Jan Leike**
DeepMind
leike@google.com

**Tobias Pohlen**
DeepMind
pohlen@google.com

**Geoffrey Irving**
OpenAI
irving@openai.com

**Shane Legg**
DeepMind
legg@google.com

**Dario Amodei**
OpenAI
damodei@openai.com

## Abstract

To solve complex real-world problems with reinforcement learning, we cannot rely on manually specified reward functions. Instead, we can have humans communicate an objective to the agent directly. In this work, we combine two approaches to learning from human feedback: expert demonstrations and trajectory preferences. We train a deep neural network to model the reward function and use its predicted reward to train an DQN-based deep reinforcement learning agent on 9 Atari games. Our approach beats the imitation learning baseline in 7 games and achieves strictly superhuman performance on 2 games without using game rewards. Additionally, we investigate the goodness of fit of the reward model, present some reward hacking problems, and study the effects of noise in the human labels.

## 1  Introduction

Reinforcement learning (RL) has recently been very successful in solving hard problems in domains with well-specified reward functions (Mnih et al., 2015, 2016; Silver et al., 2016). However, many tasks of interest involve goals that are poorly defined or hard to specify as a hard-coded reward. In those cases we can rely on demonstrations from human experts (inverse reinforcement learning, Ng and Russell, 2000; Ziebart et al., 2008), policy feedback (Knox and Stone, 2009; Warnell et al., 2017), or trajectory preferences (Wilson et al., 2012; Christiano et al., 2017).

When learning from demonstrations, a policy model is trained to *imitate* a human demonstrator on the task (Ho and Ermon, 2016; Hester et al., 2018). If the policy model mimics the human expert's behavior well, it can achieve the performance of the human on the task. However, to provide meaningful demonstrations, the human demonstrator has to have some familiarity with the task and understand how to perform it. In this sense, imitation learning puts more burden on the human than just providing feedback on behavior, which only requires the ability to judge outcomes. Moreover, using this imitation learning approach it is impossible to significantly exceed human performance.

To improve on imitation learning we can learn a reward function directly from human feedback, and optimize it using reinforcement learning. In this work, we focus on reward learning from trajectory preferences in the same way as Christiano et al. (2017). However, learning a reward function from trajectory preferences expressed by a human suffers from two problems:

1. It is hard to obtain a good state space coverage with just random exploration guided by preferences. If the state space distribution is bad, then the diversity of the trajectory that we request preferences for is low and thus the human in the loop can't convey much meaningful information to the agent.

2. Preferences are an inefficient way of soliciting information from humans, providing only a few hundred bits per hour per human.

Our approach addresses the problems in imitation learning and learning from trajectory preferences by combining the two forms of feedback. First, we initialize the agent's policy with imitation learning from the expert demonstrations using the pretraining part of the DQfD algorithm (Hester et al., 2018). Second, using trajectory preferences and expert demonstrations, we train a reward model that lets us improve on the policy learned from imitation.

We evaluate our method on the Arcade Learning Environment (Bellemare et al., 2013) because Atari games are RL problems difficult enough to benefit from nonlinear function approximation and currently among the most diverse environments for RL. Moreover, Atari games provide well-specified 'true' reward functions, which allows us to objectively evaluate the performance of our method and to do more rapid experimentation with 'synthetic' (simulated) human preferences based on the game reward.

We show that demonstrations mitigate problem 1 by allowing a human that is familiar with the task to guide exploration consistently. This allows us to learn to play exploration-heavy Atari games such as Hero, Private Eye, and Montezuma's Revenge. Moreover, in our experiments, using demonstrations typically halves the amount of human time required to achieve the same level of performance; demonstrations alleviate problem 2 by allowing the human to communicate more efficiently.

## 1.1 Related work

**Learning from human feedback.**    There is a large body of work on reinforcement learning from human ratings or rankings (Wirth et al., 2017): Knox and Stone (2009), Pilarski et al. (2011), Akrour et al. (2012), Wilson et al. (2012), Wirth and Fürnkranz (2013), Daniel et al. (2015), El Asri et al. (2016), Wirth et al. (2016), Mathewson and Pilarski (2017), and others. Focusing specifically on deep RL, Warnell et al. (2017) extend the TAMER framework to high-dimensional state spaces, using feedback to train the policy directly (instead of the reward function). Lin et al. (2017) apply deep RL from human feedback to 3D environments and improve the handling of low-quality or intermittent feedback. Saunders et al. (2018) use human feedback as a blocker for unsafe actions rather than to directly learn a policy. The direct predecessor of our work is Christiano et al. (2017), with similar tasks, rewards, policy architectures, and preference learning scheme.

**Combining imitation learning and deep RL.**    Various work focuses on combining human demonstrations with deep RL. Hester et al. (2018), on whose method this work is based, use demonstrations to pretrain a Q-function, followed by deep Q-learning with the demonstrations as an auxiliary margin loss. Večerík et al. (2017) apply the same technique to DDPG in robotics, and Zhang and Ma (2018) pretrain actor-critic architectures with demonstrations. Nair et al. (2018) combine these methods with hindsight experience replay (Andrychowicz et al., 2017). Zhu et al. (2018) combine imitation learning and RL by summing an RL loss and a generative adversarial loss from imitating the demonstrator (Ho and Ermon, 2016). Finally, the first published version of AlphaGo (Silver et al., 2016) pretrains from human demonstrations. Our work differs from all these efforts in that it replaces the hand-coded RL reward function with a learned reward function; this allows us to employ the imitation learning/RL combination even in cases where we cannot specify a reward function.

**Inverse reinforcement learning (IRL).**    IRL (Ng and Russell, 2000; Abbeel and Ng, 2004; Ziebart et al., 2008) use demonstrations to infer a reward function. Some versions of our method make use of the demonstrations to train the reward function—specifically, our autolabel experiments label the demonstrations as preferable to the agent policy. This is closely related to generative adversarial imitation learning (Ho and Ermon, 2016), a form of IRL. Note, however, that in addition to training the reward function from demonstrations we also train it from direct human feedback, which allows us to surpass the performance of the demonstrator in 2 out of 9 games.

**Reward-free learning.**    Reward-free learning attempts to avoid reward functions and instead use measures of intrinsic motivation, typically based on information theory, as a training signal (Chentanez et al., 2005; Schmidhuber, 2006; Orseau et al., 2013). The intrinsic motivation measure may include mutual information between actions and end states (Gregor et al., 2016), state prediction error or surprise (Pathak et al., 2017), state visit counts (Storck et al., 1995; Bellemare et al., 2016),

distinguishability to a decoder (Eysenbach et al., 2018), or empowerment (Salge et al., 2014), which is also related to mutual information (Mohamed and Rezende, 2015). The present work differs from reward-free learning in that it attempts to learn complex reward functions through interaction with humans, rather than replacing reward with a fixed intrinsic objective.

## 2   Method

### 2.1   Setting

We consider an agent that is interacting sequentially with an environment over a number of time steps (Sutton and Barto, 2018): in time step $t$ the agent receives an observation $o_t$ from the environment and takes an action $a_t$. We consider the episodic setting in which the agent continues to interact until a terminal time step $T$ is reached and the episode ends. Then a new episode starts. A *trajectory* consists of the sequence $(o_1, a_1), \ldots (o_T, a_T)$ of observation-action pairs.

Typically in RL the agent also receives a reward $r_t \in \mathbb{R}$ at each time step. Importantly, in this work we are not assuming that such reward is available directly. Instead, we assume that there is a human in the loop who has an intention for the agent's task, and communicates this intention to the agent using two feedback channels:

1. *Demonstrations*: several trajectories of human behavior on the task.

2. *Preferences*: the human compares pairwise short trajectory segments of the agent's behavior and prefers those that are closer to the intended goal (Christiano et al., 2017).

In our setting, the demonstrations are available from the beginning of the experiment, while the preferences are collected during the experiment while the agent is training.

The goal of the agent is to approximate as closely as possible the behavior intended by the human. It achieves this by 1. imitating the behavior from the demonstrations, and 2. attempting to maximize a reward function inferred from the preferences and demonstrations. This is explained in detail in the following sections.

### 2.2   The training protocol

Our method for training the agent has the following components: an *expert* who provides demonstrations; an *annotator* (possibly the same as the expert) who gives preference feedback; a *reward model* that estimates a reward function from the annotator's preferences and, possibly, the demonstrations; and the *policy*, trained from the demonstrations and the reward provided by the reward model. The reward model and the policy are trained jointly according to the following protocol:

---
**Algorithm 1** Training protocol
---
1: The expert provides a set of demonstrations.
2: Pretrain the policy on the demonstrations using behavioral cloning using loss $J_E$.
3: Run the policy in the environment and store these 'initial trajectories.'
4: Sample pairs of clips (short trajectory segments) from the initial trajectories.
5: The annotator labels the pairs of clips, which get added to an annotation buffer.
6: (Optionally) automatically generate annotated pairs of clips from the demonstrations and add them to the annotation buffer.
7: Train the reward model from the annotation buffer.
8: Pretrain of the policy on the demonstrations, with rewards from the reward model.
9: **for** $M$ iterations **do**
10:     Train the policy in the environment for $N$ steps with reward from the reward model.
11:     Select pairs of clips from the resulting trajectories.
12:     The annotator labels the pairs of clips, which get added to the annotation buffer.
13:     Train the reward model for $k$ batches from the annotation buffer.
14: **end for**
---

Note that we pretrain the policy model twice before the main loop begins. The first pretraining is necessary to elicit preferences for the reward model. The policy is pretrained again because some components of the DQfD loss function require reward labels on the demonstrations (see next section).

## 2.3 Training the policy

The algorithm we choose for reinforcement learning with expert demonstrations is deep Q-Learning from demonstrations (DQfD; Hester et al., 2018), which builds upon DQN (Mnih et al., 2015) and some of its extensions (Schaul et al., 2015; Wang et al., 2016; Hasselt et al., 2016). The agent learns an estimate of the action-value function (Sutton and Barto, 2018) $Q(o, a)$, approximated by a deep neural network with parameters $\theta$ that outputs a set of action-values $Q(o, \cdot; \theta)$ for a given input observation $o$. This action-value function is learned from demonstrations and from agent experience, both stored in a replay buffer (Mnih et al., 2015) in the form of transitions $(o_t, a_t, \gamma_{t+1}, o_{t+1})$, where $\gamma$ is the reward discount factor (fixed value at every step except 0 at end of an episode). Note that the transition does not include the reward, which is computed from $o_t$ by the reward model $\hat{r}$.

During the pretraining phase, the replay buffer contains only the transitions from expert demonstrations. During training, agent experience is added to the replay buffer. The buffer has a fixed maximum size, and once it is full the oldest transitions are removed in a first-in first-out manner. Expert transitions are always kept in the buffer. Transitions are sampled for learning with probability proportional to a priority, computed from their TD error at the moment they are added to and sampled from the buffer (Schaul et al., 2015).

The training objective for the agent's policy is the the cost function $J(Q) = J_{PDDQn}(Q) + \lambda_2 J_E(Q) + \lambda_3 J_{L2}(Q)$. The term $J_{PDDQn}$ is the prioritized (Schaul et al., 2015) dueling (Wang et al., 2016) double (Hasselt et al., 2016) Q-loss (PDD), combining 1- and 3-step returns (Hester et al., 2018). This term attempts to ensure that the $Q$ values satisfy the Bellman equation (Sutton and Barto, 2018). The term $J_E$ is a large-margin supervised loss, applied only to expert demonstrations. This term tries to ensure that the value of the expert actions is above the value of the non-expert actions by a given margin. Finally, the term $J_{L2}$ is an $L2$-regularization term on the network parameters. The hyperparameters $\lambda_2$ and $\lambda_3$ are scalar constants. The agent's behavior is $\epsilon$-greedy with respect to the action-value function $Q(o, \cdot; \theta)$.

## 2.4 Training the reward model

Our reward model is a convolutional neural network $\hat{r}$ taking observation $o_t$ as input (we omit actions in our experiments) and outputting an estimate of the corresponding reward $r_{t+1} \in \mathbb{R}$. Since we do not assume to have access to an environment reward, we resort to indirect training of this model via preferences expressed by the annotator (Christiano et al., 2017). The annotator is given a pair of *clips*, which are trajectory segments of 25 agent steps each (approximately 1.7 seconds long). The annotator then indicates which clip is preferred, that the two clips are equally preferred, or that the clips cannot be compared. In the latter case, the pair of clips is discarded. Otherwise the judgment is recorded in an annotation buffer $A$ as a triple $(\sigma^1, \sigma^2, \mu)$, where $\sigma^1, \sigma^2$ are the two episode segments and $\mu$ is the judgment label (one of $(0, 1)$, $(1, 0)$ or $(0.5, 0.5)$).

To train the reward model $\hat{r}$ on preferences, we interpret the reward model as a preference predictor by assuming that the annotator's probability of preferring a segment $\sigma^i$ depends exponentially on the value of the reward summed over the length of the segment:

$$\hat{P}[\sigma^1 \succ \sigma^2] = \exp\left(\sum_{o \in \sigma^1} \hat{r}(o)\right) \Big/ \left(\exp\left(\sum_{o \in \sigma^1} \hat{r}(o)\right) + \exp\left(\sum_{o \in \sigma^2} \hat{r}(o)\right)\right)$$

We train $\hat{r}$ to minimize the cross-entropy loss between these predictions and the actual judgment labels:

$$\text{loss}(\hat{r}) = - \sum_{(\sigma^1, \sigma^2, \mu) \in A} \mu(1) \log \hat{P}[\sigma^1 \succ \sigma^2] + \mu(2) \log \hat{P}[\sigma^2 \succ \sigma^1]$$

This follows the Bradley-Terry model (Bradley and Terry, 1952) for estimating score functions from pairwise preferences. It can be interpreted as equating rewards with a preference ranking scale analogous to the Elo ranking system developed for chess (Elo, 1978).

Since the training set is relatively small (a few thousand pairs of clips) we incorporate a number of modifications to prevent overfitting: adaptive regularization, Gaussian noise on the input, L2 regularization on the output (details in Appendix A). Finally, since the reward model is trained only on comparisons, its scale is arbitrary, and we normalize it every 100,000 agent steps to be zero-mean and have standard deviation 0.05 over the annotation buffer $A$. This value for the standard deviation was chosen empirically; deep RL is very sensitive to the reward scale and this parameter is important for the stability of training.

### 2.5 Selecting and annotating the video clips

The clips for annotation are chosen uniformly at random from the initial trajectories (line 3 in Algorithm 1) and the trajectories generated during each iteration of the training protocol. Ideally we would select clips based on uncertainty estimates from the reward model; however, the ensemble-based uncertainty estimates used by Christiano et al. (2017) did not improve on uniform sampling and slowed down the reward model updates. The annotated pairs are added to the annotation buffer, which stores all the pairs that have been annotated so far. The number of pairs collected after each protocol iteration decreases as the experiment progresses, according to a schedule (see details in Appendix A).

In some experiments we attempt to leverage the expert demonstrations to enrich the set of initial labels. In particular, each clip selected for annotation from the initial trajectories is paired with a clip selected uniformly at random from the demonstrations and a labeled pair is automatically generated in which the demonstration is preferred. Thus the initial batch of $k$ pairs of clips produces $2k$ extra annotated pairs without invoking the annotator, where $k$ is the number of labels initially requested from the annotator.

In the majority of our experiments the annotator is not a human. Instead we use a synthetic oracle whose preferences over clips reflect the true reward of the underlying Atari game. This *synthetic feedback* allows us to run a large number of simulations and investigate the quality of the learned reward in some detail (see Section 3.2).

## 3 Experimental results

Our goal is to train an agent to play Atari games *without access to the game's reward function*. Therefore typical approaches, such as deep RL (Mnih et al., 2015, 2016) and deep RL with demos (Hester et al., 2018) cannot be applied here. We compare the following experimental setups (details are provided in Appendix A):

1. *Imitation learning* (first baseline). Learning purely from the demonstrations without reinforcement learning (Hester et al., 2018). In this setup, no preference feedback is provided to the agent.

2. *No demos* (second baseline). Learning from preferences without expert demonstrations, using the setup from Christiano et al. (2017) with PDD DQN instead of A3C.

3. *Demos + preferences*. Learning from both preferences *and* expert demonstrations.

4. *Demos + preferences + autolabels*. Learning from preferences and expert demonstrations, with additional preferences automatically gathered by preferring demo clips to clips from the initial trajectories (see Section 2.5).

We've selected 9 Atari games, 6 of which (Beamrider, Breakout, Enduro, Pong, Q*bert, and Seaquest) feature in Mnih et al. (2013) and Christiano et al. (2017). Compared to previous work we exclude Space Invaders because we do not have demonstrations for it. The three additional games (Hero, Montezuma's Revenge, and Private Eye) were chosen for their exploration difficulty: without the help of demonstrations, it is very hard to perform well in them (Hester et al., 2018).

In each experimental setup (except for imitation learning) we compare four feedback schedules. The full schedule consists of 6800 labels (500 initial and 6300 spread along the training protocol). The other three schedules reduce the total amount of feedback by a factor of 2, 4 and 6 respectively (see details in Appendix A).

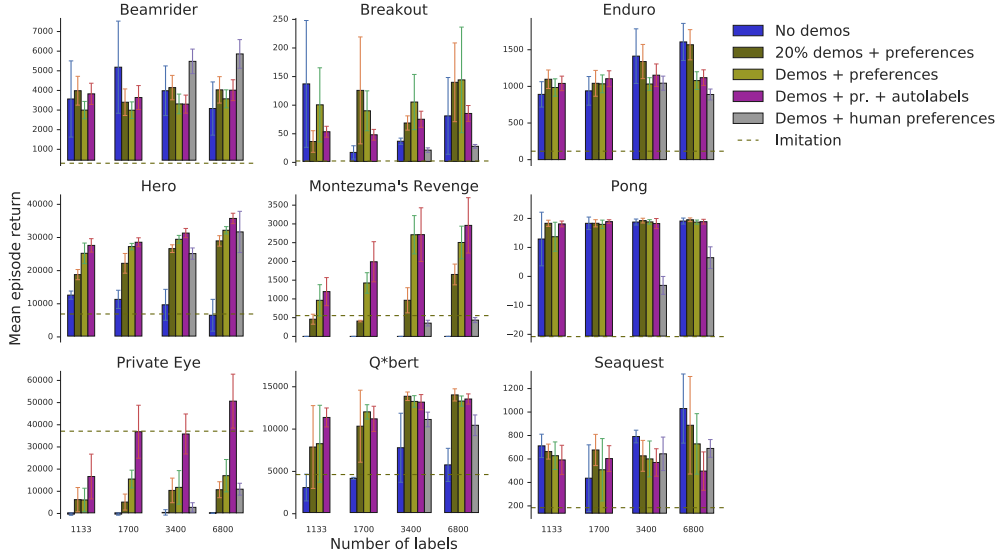

Figure 1: Performance of our method on 9 Atari games after 50 million agent steps, for different annotation schedules and training setups: *no demos* is the reward learning setup used by Christiano et al. (2017), trained with DQN; *imitation* is the baseline from DQfD without RL; *demos + preferences* and *demos + pr. + autolables* use all demos and synthetic labels, with and without automatic labels from demos; *20% demos + preferences* is like *demos + preferences* but uses only 20% of the available demos; *demos + human preferences* is the same setup as *demos + preferences*, but with a human instead of the synthetic oracle. The vertical lines depict the standard deviation across three runs of each experiment.

The majority of the experiments use the synthetic oracle for labeling. We also run experiments with actual human annotators in the *demos + preferences* experimental setup, with the full schedule and with the schedule reduced by a factor of 2. In our experiments the humans were contractors with no experience in RL who were instructed as in Christiano et al. (2017) to only judge the outcome visible in the segments. We label these experiments as *human*.

Figure 1 displays the mean episode returns in each game, setup and schedule, after 50 million agent steps. We can compare the relative performance across four different experimental setups:

*How much do preferences help (demos + preferences vs. imitation)?* Our approach outperforms the imitation learning baseline in all games except Private Eye. In 6 of the 9 games this holds in every condition, even with the smallest amount of feedback. The bad performance of imitation learning in most Atari tasks is a known problem (Hester et al., 2018) and in the absence of a reward function preference feedback offers an excellent complement. Private Eye is a stark exception: imitation is hard to beat even with access to reward (Hester et al., 2018), and in our setting preference feedback is seriously damaging, except when the demonstrations themselves are leveraged for labeling.

*How much do demos help (demos + preferences vs. no demos)?* Hero, Montezuma's Revenge, Private Eye and Q*bert benefit greatly from demonstrations. Specifically, in Montezuma's Revenge and Private Eye there is no progress solely from preference feedback; without demonstrations Hero does not benefit from increased feedback; and in Q*bert demonstrations allow the agent to achieve better performance with the shortest label schedule (1100 labels) than with the full no-demos schedule. With just 20% of the demonstrations (typically a single episode) performance already improves significantly[1]. In the rest of the games the contribution of demonstrations is not significant, except for Enduro, where it is harmful, and possibly Seaquest. In Enduro this can be explained by the relatively poor performance of the expert: this is the only game where the trained agents are superhuman in all conditions. Note that our results for *no demos* are significantly different from those in Christiano

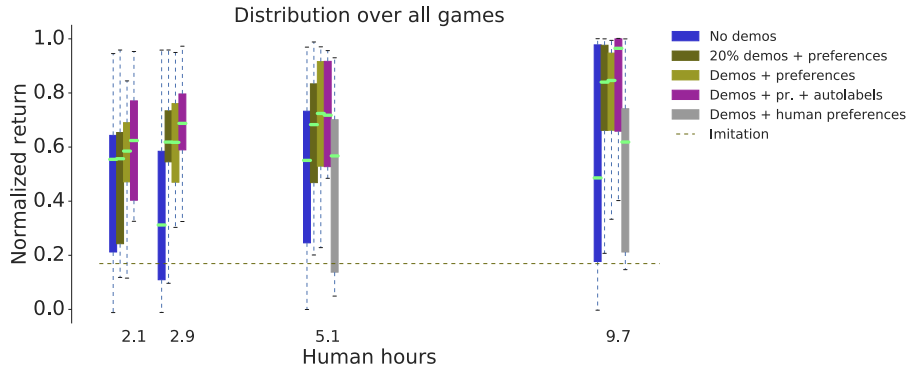

Figure 2: Aggregated performance over all games after 50 million agent steps for different schedules and training setups. Performance is normalized for each game between 0 (return of a random policy) and 1 (best return across all setups and schedules). The boxplots show the distribution over all 9 games, the bright notch representing the median, boxes reaching the 25 and 75 percentiles, and whiskers the whole range. Their position along the $x$ axis shows with the total number of annotation labels used.

et al. (2017) because we use DQN (Mnih et al., 2015) instead of A3C (Mnih et al., 2016) to optimize the policy (see Appendix F).

*How does human feedback differ from the synthetic oracle (demos + preferences vs. human)?* Only in Beamrider is human feedback superior to synthetic feedback (probably because of implicit reward shaping by the human). In most games performance is similar, but in Breakout, Montezuma's Revenge and Pong it is clearly inferior. This is due to attempts at reward shaping that produce misaligned reward models (see Figure 3 and Appendix D) and, in the case of Montezuma's Revenge, to the high sensitivity of this game to errors in labeling (see Appendix E).

*How much do automatic preference labels help (demos + preference vs. demos + preferences + auto labels)?* Preference labels generated automatically from demonstrations increase performance in Private Eye, Hero, and Montezuma's Revenge, where exploration is difficult. On most games, there are no significant differences, except in Breakout where human demonstrations are low quality (they do not 'tunnel behind the wall') and thus hurt performance.

## 3.1 Use of human time

Figure 2 summarizes the overall performance of each setup by human time invested. More than half of the games achieve the best performance with full feedback and the help of demonstrations for imitation and annotation, and, for each feedback schedule, the majority of games benefit from demonstrations, and from the use of demonstrations in annotation. With only 3400 labels even the worst-performing game with demonstrations and automatic labels beats the median performance without demonstrations and the full 6800 labels. If demonstrations are not available there are games that never go beyond random-agent scores; demonstrations ensure a minimum of performance in any game, as long as they are aided by some preference feedback. For further details refer to Appendix B.

## 3.2 Quality of reward model

In our experiments we are evaluating the agent on the Atari game score, which may or may not align with the reward from the reward model that the agent is trained on. With synthetic labels the learned reward should be a good surrogate of the true reward, and bad performance can stem from two causes: (1) failure of the reward model to fit the data, or (2) failure of the agent to maximize the learned reward. With human labels there are two additional sources of error: (3) mislabeling and (4) a misalignment between the true (Atari) reward function and the human's reward function. In this section we disentangle these possibilities.

Learning the reward model is a supervised learning task, and in Appendix C we argue that it succeeds in fitting the data well. Figure 3 compares the learned reward model with the true reward in three

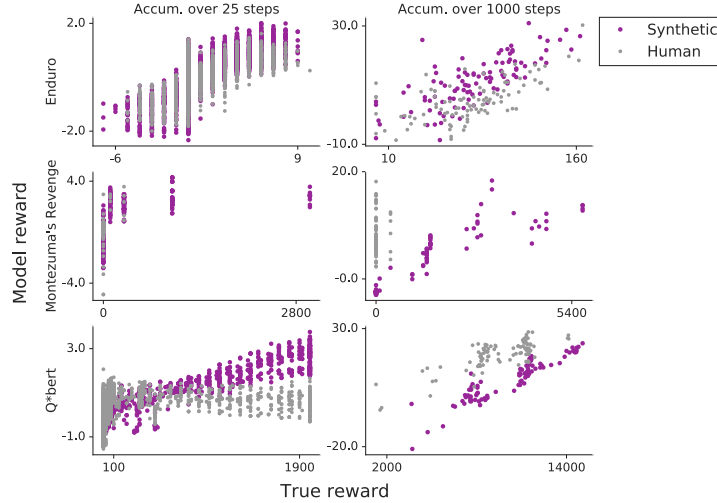

Figure 3: True vs. learned reward accumulated in sequences of 25 (left) and 1000 (right) agent steps in Enduro, Montezuma's Revenge and Q*bert. Magenta and gray dots represent the model learned from synthetic and human preferences, respectively. A fully aligned reward model would have all points on a straight line. For this evaluation, the agent policy and reward model were fixed after successful full-schedule training (in the case of synthetic preference feedback we chose the most successful seed; in the case of human preference feedback only one run was available).

games (see Appendix D for the other six games). Both synthetic (*demos + pr. + autolabels* in Figure 1) and human preference models are presented for comparison. Perfect alignment between true reward and modelled reward is achieved if they are equal up to an affine-linear transformation; in this case all points in the plot would be on a straight line. In most games the synthetically trained reward model is reasonably well-aligned, so we can rule out cause (1).

In Enduro both human and synthetic preferences produce well-aligned reward models, especially over long time horizons. Q*bert presents an interesting difference between human and synthetic preferences: on short timescales, the human feedback does not capture fine-grained reward distinctions (e.g., whether the agent covered one or two tiles) which are captured by the synthetic feedback. However, on long timescales this does not matter much and both models align well. A similar pattern occurs in Hero. Finally, in Montezuma's Revenge human feedback fails while synthetic feedback succeeds. This is partially due to a misalignment (because the human penalizes death while the Atari score does not) and partially due to the sensitivity of the reward function to label noise. For more details, see Appendix D.

The difference between synthetically and human-trained reward model captures causes (3) and (4). To disentangle (3) and (4), we also provide experiments with a mislabeling rate in Appendix E.

**Reward hacking.** To further evaluate the quality of the reward model, we run experiments with frozen reward models obtained from successful runs. The result is shown in Figure 4, left. Although a fully trained model should make learning the task easier, in no case is the fixed-model performance significantly better than the online training performance, which suggests that joint training of agent and reward is not intrinsically problematic. Moreover, in Hero, Montezuma, and Private Eye the performance with a fully trained reward model is much worse than online reward model training. In these cases the drop in performance happens when the agent learns to exploit undesired loopholes in the reward function (Figure 4, right), dramatically increasing the predicted reward with behaviors that diminish the true score.[2] These loopholes can be fixed interactively when the model is trained online with the agent, since exploitative behaviors that do not lead to good scores can be annotated as soon as they feature significantly in the agent's policy, similar to adversarial training (Goodfellow et al., 2014). With online training we also observed cases where performance temporarily drops, with simultaneous increases in model reward, especially when labels are noisy (Appendix E).

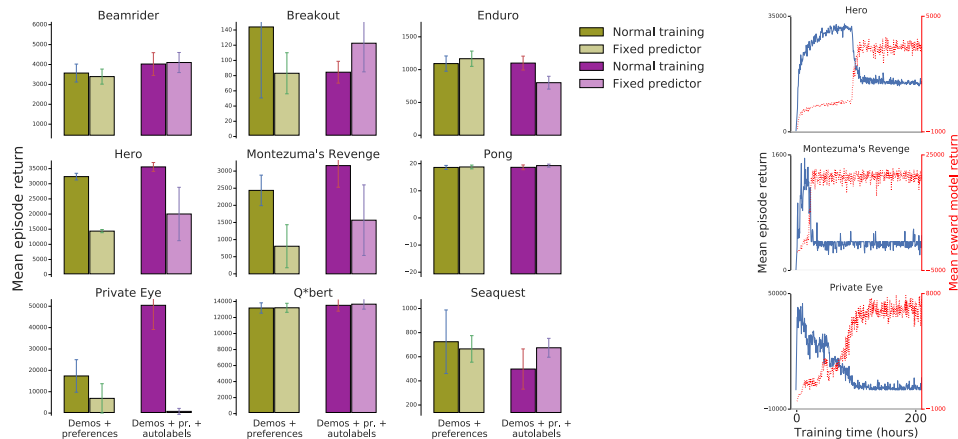

Figure 4: Failure modes when training from a frozen reward model (contrary to our method). Left: performance at each game after 50 million agent steps. The darker colored bars show the results from our training protocol (same as Figure 1) with the full label schedule. The reward model from the best seed in these experiments is then frozen and used to train an agent from scratch, resulting in the lighter colored bars. Right: average true return (blue) and average reward model return (red) during training of three games (only one seed shown per game) from a frozen reward model. This showcases how the agent learns to exploit the reward model: over time the perceived performance (according to the reward model) increases, while the actual performance (according to the game score) plummets.

## 4 Discussion

Combining both preferences and demonstrations outperforms using either in isolation. Their combination is an effective way to provide guidance to an agent in the absence of explicit reward (Figure 1). Even small amounts of preference feedback (about 1000 comparisons) let us outperform imitation learning in 7 out of 9 games. Moreover, the addition of demonstrations to learning from preferences typically results in substantial performance gains, especially in exploration-heavy games. We achieve superhuman performance on Pong and Enduro, which is impossible even with perfect imitation.

Synthetic preference feedback proved more effective than feedback provided by humans. It could be expected that human feedback has the advantage in the exploration-heavy games, where the human can shape the reward to encourage promising exploration strategies. Analysis of the labels shows that the human annotator prefers clips where the agent seems to be exploring in particular directions. However, instead of encouraging exploration, this feedback produces 'reward pits' that trap the agent into repetitive and fruitless behaviors. This effect is not novel; MacGlashan et al. (2017) have previously argued that humans are bad at shaping reward. However, our results show that demonstrations can provide consistent exploration guidance.

In addition to the experiments presented here, we were unsuccessful at achieving significant performance improvements from a variety of other ideas: distributional RL (Bellemare et al., 2017), quantile distributional RL (Dabney et al., 2017), weight sharing between reward model and policy, supplying the actions as input to the reward model, pretrained convolutional layers or semi-supervised training of the reward model, phasing out of the large-margin supervised loss along training, and other strategies of annotation from demos (see Appendix H).

In contrast to Christiano et al. (2017), whose work we build upon here, we use the value-based agent DQN/DQfD instead of the policy-gradient-based agent A3C. This shows that learning reward functions is feasible across two very different RL algorithms with comparable success. Appendix F compares the scores of the two agents.

Finally, Section 3.2 highlights a caveat of reward learning: sometimes the agent learns to exploit unexpected sources of reward. This so-called *reward hacking* problem (Amodei et al., 2016; Everitt, 2018) is not unique to reward learning; hard-coded reward functions are also exploitable in this way (Lehman et al., 2018). Importantly, we only found persistent reward hacking when the preference feedback was frozen. This suggests that our method, keeping a human in the training loop who provides *online* feedback to the agent, is effective in preventing reward hacking in Atari games.

**Acknowledgements**

We thank Serkan Cabi, Bilal Piot, Olivier Pietquin, Tom Everitt, and Miljan Martic for helpful feedback and discussions. Moreover, we thank Elizabeth Barnes for proofreading the paper and Ashwin Kakarla, Ethel Morgan, and Yannis Assael for helping us set up the human experiments. Last but not least, we are grateful to the feedback annotators for their many hours of meticulous work.

## Footnotes

[1]Experiments with 50% of the demonstrations (not shown) produced scores similar to the full demo experiments—the benefits of demonstration feedback seem to saturate quickly.

[2]Videos at `https://youtube.com/playlist?list=PLehfUY5AEKX-g-QNM7FsxRHgiTOC1-1hv`

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
