[Supplementary Material]

# A Experimental details

## A.1 Environment

We use the Arcade Learning Environment (Bellemare et al., 2013) with the standard set of environment wrappers used by Mnih et al. (2015): full set of 18 actions, 0 to 90 no-ops in the beginning of an episode, max-pooling over adjacent frames with action repeat and frame stacking of 4 frames, observation resizing to 84x84 and converting to grayscale. Across this paper we treat 4 frames as one observation with action repeat as a single actor step, i.e., one actor step corresponds to 4 Atari frames.

We replace the score with a constant black background to prevent inferring the reward from the score. Life loss and end-of-episode signals are not passed to the agent, effectively converting the environment into a single continuous episode. When providing synthetic oracle feedback we replace episode ends with a penalty in all games except Pong; the agent must learn this penalty.

## A.2 Expert demonstrations

The same set of demonstrations for each game is used in all experiments. Our demonstrations are the same demonstrations as used by Hester et al. (2018) and were collected from an expert game tester. The length and scores of these demonstrations are as follows.

| Game | Episodes | Transitions | Avg score | Min score | Max score |
|---|---|---|---|---|---|
| Beamrider | 4 | 38665 | 16204 | 12594 | 19844 |
| Breakout | 9 | 10475 | 38 | 17 | 79 |
| Enduro | 5 | 42058 | 641 | 383 | 803 |
| Hero | 5 | 32907 | 71023 | 35155 | 99320 |
| Montezuma's Revenge | 5 | 17949 | 33220 | 32300 | 34900 |
| Pong | 3 | 17719 | -8 | -12 | 0 |
| Private Eye | 5 | 10899 | 72415 | 70375 | 74456 |
| Q*bert | 5 | 75472 | 89210 | 80700 | 99450 |
| Seaquest | 7 | 57453 | 74374 | 56510 | 101120 |

## A.3 Agent and reward model

We optimize policies using the DQfD algorithm (Hester et al., 2018), with standard architecture and parameters: dueling, double Q-learning with a target network updated every 8000 actor steps, discount $\gamma = 0.99$, a mix of 1- and 3-step returns, prioritized replay (Schaul et al., 2015) based on TD error with exponent $\alpha = 0.5$ and importance sampling exponent $\beta = 0.4$. The buffer size of $1e6$ for actor experience plus permanent demonstrations, batch size 32, learning update every 4 steps, additional large-margin supervised loss for expert demonstrations (Q margin=1, loss weight=1), priority bonus for expert demonstrations $\epsilon_d = 3$. We stack 4 steps as input to the Q-value network. The optimizer is Adam (Kingma and Ba, 2014) with learning rate 0.0000625, $\beta_1 = 0.9$, $\beta_2 = 0.999$, $\epsilon = 0.00015625$. Importantly, each time a batch is sampled from the buffer for learning, the reward values corresponding to the batch are computed using the reward model.

The DQfD agent policy is $\epsilon$-greedy with epsilon annealed linearly from 0.1 to 0.01 during the first $10^5$ actor steps.

For the reward model, we use the same configuration as the Atari experiments in Christiano et al. (2017): 84x84x4 stacked frames (same as the inputs to the policy) as inputs to 4 convolutional layers of size 7x7, 5x5, 3x3, and 3x3 with strides 3, 2, 1, 1, each having 16 filters, with leaky ReLU nonlinearities ($\alpha = 0.01$). This is followed by a fully connected layer of size 64 and then a scalar output. The agent action $a_t$ is not used as input as this did not improve performance. Since the training set is relatively small (a few thousand pairs of clips) we incorporate a number of modifications to this basic approach to prevent overfitting:

- A fraction of $1/e$ of the data is held out to be used as a validation set. We use L2-regularization of network weights with the adaptive scheme described in Christiano et al. (2017): the L2-regularization weight increases if the average validation loss is more than 50% higher than the average training loss, and decreases if it is less than 10% higher (initial weight 0.0001, multiplicative rate of change 0.001 per learning step).

- An extra loss proportional to the square of the predicted rewards is added to impose a zero-mean Gaussian prior on the reward distribution.
- Gaussian noise of amplitude $0.1$ (the grayscale range is 0 to 1) is added to the inputs.
- Convolutional layers use batch normalization (Ioffe and Szegedy, 2015) with decay rate $0.99$ and per-channel dropout (Srivastava et al., 2014) with $\alpha = 0.8$.
- We assume there is a 10% chance that the annotator responds uniformly at random, so that the model will not overfit to possibly erroneous preferences. We account for this error rate by using $\hat{P}_e = 0.9\hat{P} + 0.05$ instead of $\hat{P}$ for the cross-entropy computation.

Finally, since the reward model is trained merely on comparisons, its absolute scale is arbitrary. Therefore we normalize its output so that it has 0 mean and standard deviation $0.05$ across the annotation buffer. We do *not* use an ensemble of reward models as done by Christiano et al. (2017). The model is trained on batches of 16 segment pairs (see below), optimized with Adam (Kingma and Ba, 2014) with learning rate $0.0003$, $\beta_1 = 0.9$, $\beta_2 = 0.999$, and $\epsilon = 10^{-8}$.

The training set for the reward model (the annotation buffer) consists of annotated pairs of clips, each of 25 actor steps (1.7 seconds long at 15 fps with frame skipping). The size of the training set grows over the course of the experiment as segments are collected according to an annotation schedule. The 'full' annotation schedule consists of a set of 500 labels from initial trajectories at the beginning of training, followed by a decreasing rate of annotation, roughly proportional to $5 \cdot 10^6/(T + 5 \cdot 10^6)$, where $T$ is the number of actor steps. The total number of labels in a 50 million-step experiment is 6800. We compare performance with proportionally reduced schedules that have 2, 4 and 6 times fewer labels than the full schedule (including the initial batch), with a total number of labels of 3400, 1700 and 1133 respectively.

## A.4  Training protocol

The training protocol consists of 500 iterations and each iteration consists of $10^5$ agent steps. The reward model is fixed during each iteration. Trajectories (effectively one long episode per iteration, since we removed episode boundaries) are collected in each iteration and clips of 25 agent steps are picked at random for annotation. When using synthetic annotation, clip pairs are labeled and added to the annotation set immediately after each iteration. In experiments with preference feedback from real human annotators, clips are labeled about every 6 hours, corresponding to about 12 iterations. After each iteration, the reward model is trained for 6250 batches sampled uniformly from the annotation buffer.

A pretraining phase precedes the training iterations. The pretraining phase consists of the following:

- Training the agent purely from demonstrations. This includes both the imitation large-margin loss and the Q loss from expert demonstrations. Notice that, since the reward model has not yet been trained at this point, it predicts small random values, and the Q loss, which is based on those predicted rewards, is noisy and acts as a regularizer. In this phase of pretraining we train on $20,000$ batches.
- Generating trajectories from the policy trained in the previous steps to collect the initial set of 500 clip pairs (250, 125 or 83 in the case of reduced schedules).
- Labeling the initial set of clip pairs. In some experiments, an additional set of 1000 (500, 250, 167 for reduced schedules) labeled pairs is automatically generated by comparing each clip in each pair of the initial set with a clip sampled uniformly at random from the demonstrations. It is automatically labeled to prefer the clip from the demonstration.
- Training the reward model with 50000 batches from the labeled clips.
- Another round of agent training purely from demonstrations. Unlike the first pretraining phase, the reward model has now undergone some training and the Q loss is more meaningful. This last phase of pretraining consists of 60000 batches.

# B  Performance as a function of human effort

Figure 5 shows the performance in each game as a function of joint labeling and demonstration effort (measured in human hours), for the different preference feedback schedules (full 6800-sample

Figure 5: Performance at each game as a function of human (or synthetic) effort, adding labeling time (at 750 labels/hour) and demonstration time (at 15 fps).

Figure 6: Best performance up to a given human effort (in terms of time) for each game, with and without demonstrations. The different colors of the with-demos lines correspond to the best setup, as displayed in Figure 5

schedule, and 1/2, 1/4 and 1/6 thereof) and learning setups (no demonstrations, demonstrations + preferences, demonstrations + preferences + initial automatic labels from demonstrations, and demonstrations with non-synthetic, actual human preferences). This information is synthesized in Figure 6, where the best achievable performance for a given amount of effort is displayed, either in the no-demonstration setup or in setups that make use of demonstrations.

The more demo-driven games (Hero, Montezuma's Revenge, Private Eye and Q*bert) are also more feedback-driven, improving with additional feedback if demonstrations are available. The opposite is true for Enduro, where the score increases with additional feedback only if demonstrations are not used. The agent easily beats the demonstrations in this game, so they work against the feedback. Pong is solved with very little synthetic feedback in any setup, but when preferences are provided by a human it significantly improves with extra feedback. Beamrider without demonstrations peaks at low feedback and then regresses, but with demonstrations it is feedback-driven. Breakout and Seaquest don't display a clear trend. Breakout is especially noisy (see variance in Figure 1), the score depending greatly on whether the agent discovers 'tunneling behind the wall', absent in the demonstrations. In Seaquest the agent does not learn to resurface, limiting the scores in all setups. The reason for this is the removal of episode boundaries: the small penalty we add at the end of a game is offset by the extra time the agent can spend shooting enemies if it does not resurface. Furthermore, preferences fail to accurately capture the difference in reward magnitude between collecting all divers and bringing them to the surface compared to shooting enemies.

## C    Reward model training

Figure 7: Cross-entropy loss of the reward model (solid blue line) and average label entropy (dotted black line) during training. The average label entropy is a lower bound for the loss and depends on the ratio of 'indifferent' labels in the annotated comparisons: more 'indifferent' labels result in higher entropy.

Figure 7 depicts the reward model training loss, which is the cross-entropy between the two-class labels from clip comparison and the model output. The labels can be either $(1, 0)$ when the first clip is preferred, $(0, 1)$ when the second clip is preferred, or $(0.5, 0.5)$ when neither clip is preferred ('indifferent' label). 'Indifferent' labels have minimum possible cross-entropy $\log 2 = 0.693$, so the loss has a lower bound that depends solely on the ratio of 'indifferent' labels in the annotation dataset. This ratio varies significantly from game to game. Games with sparse rewards, like Private Eye and Montezuma's Revenge, have a high proportion of indifferent clip pairs (both segments lacking any reward) and therefore high lower bounds. The bound evolves during training as more labels are collected.

We expect a well-trained reward model to stay close to the label entropy bound. This is to some extent what happens with Beamrider, Breakout, Montezuma's Revenge and Private Eye. In other games like Hero, Q*bert and Seaquest the loss is between $50\%$ and $100\%$ above the entropy bound. The game with seemingly worst reward model training is Enduro, where the loss is more than $4$ times the lower bound. This big gap, however, can be explained by the fine-grained scoring of the game. Points are earned continuously at a rate proportional to the car speed, so very similar clips can differ by one point and not be labeled 'indifferent' by the synthetic annotator. The reward model does not latch to these preferences for similar clips, but learns to distill the important events, such as crashes that significantly reduce the accumulated score. As a result, the predicted reward is highly correlated

with the game score and the agent learns to play the game well, even if the model fails to tease apart many pairs of clips with small score differences.

# D    Reward model alignment

Figure 8: True vs. model reward accumulated in sequences of 25 (left) and 1000 (right) agent steps. Magenta and gray dots represent the model learned from synthetic (demos + pr. + autolabels in Figure 1) and human preferences, respectively. A fully aligned reward model would have all points on a straight line. For this evaluation, the agent policy and reward model were fixed after successful full-schedule training (in the case of synthetic feedback we chose the most successful seed; in the case of human feedback, there was no choice, only one run was available).

Figure 8 displays model reward plotted against true reward, accumulated over short (25 agent steps) and long (1000 agent steps) time intervals, for experiments with synthetic preference feedback and human preference feedback. With synthetic preferences, the reward model generally aligns well with the true reward, especially over longer time intervals. We can observe the following:

- Sparse rewards (as in Montezuma's Revenge and Private Eye) make preference learning harder because they result in fewer informative preference labels.
- Learning different reward sizes with preferences is hard because preferences do not express precise numeric feedback. A reward of 10 vs. 11 generates the same label as 10 vs. 100; to learn the difference the model needs to chance upon pairs of clips linked by the intermediate reward.
- The coarser alignment over short intervals makes the learned model hackable in games where imprecisely timed rewards can be exploited (see Figure 4).

As for human preferences, the reward model fails to align with the true reward in Breakout, Montezuma's Revenge and Pong. One reason is that the reward function the human wants to teach can be different from the Atari simulator's. For example, in Montezuma's Revenge the human tries to shape the reward by punishing deaths (which are not punished according to the true reward), resulting in a passive policy. In Pong and Breakout, the human gives priority to hitting the ball rather than scoring points, so sometimes it is worth dropping the ball to score an easy hit on restart.

# E    Effects of label noise

The relatively poor performance of human compared to synthetic annotators can be partially explained by random mistakes in labeling. Figure 9 shows how the different games are affected by labeling mistakes. The mistake rates observed in our human-annotated experiments are between 5% and 10%. Those noise levels have minor impact in most games, but they are significantly detrimental in Montezuma's Revenge, and thus partially accounts for the poor results in the human-labeled experiments.

Figure 9: Performance in each game with different rates of mislabelling by the annotator. Experiments are from synthetic feedback with full label schedule and not using automatic labels from demonstrations (same setting as 'human' experiments).

## F   Comparison with policy gradient

Christiano et al. (2017) used the policy-gradient-based agent A3C to evaluate preference feedback (without demonstrations). In this paper we used the value-based agent DQN/DQfD. The following table compares our scores without demonstrations with corresponding scores in Christiano et al. (2017).

| | **DQN + preferences (ours)** | | **A3C + preferences** | |
|---|---|---|---|---|
| Game | 6800 labels | 3400 labels | 5500 labels | 3300 labels |
| Beamrider | 3000 | 4000 | **10000** | **10000** |
| Breakout | **100** | 40 | 20 | 20 |
| Enduro | **1600** | 1400 | 0 | 0 |
| Pong | 19 | 19 | **20** | **20** |
| Q*bert | 5800 | 7800 | **13000** | 5000 |
| Seaquest | 1000 | 800 | **1200** | 800 |

## G   Comparison with DQfD trained from true reward

The following table compares the average scores of DQfD (Hester et al., 2018) trained from true reward and from a learned reward model (ours). Our scores are from full schedule runs with autolabels (magenta bars in Figure 1).

| Game | DQfD + feedback (ours) | DQfD + true reward | Game | DQfD + feedback (ours) | DQfD + true reward |
|---|---|---|---|---|---|
| Beamrider | 4100 | **5170** | Pong | **19** | 11 |
| Breakout | 85 | **310** | Private Eye | **52000** | 42500 |
| Enduro | 1200 | **1930** | Q*bert | 14000 | **21800** |
| Hero | 35000 | **106000** | Seaquest | 500 | **12400** |
| Montezuma's | 3000 | **4640** | | | |

Note that this does not compare like with like: while training with a synthetic oracle makes the true reward function indirectly available to the agent, our method only uses a very limited number of preference labels (feedback on at most $340.000$ agent steps), providing reward feedback on $< 1\%$ of the agent's experience. To make a fair comparison with DQfD, we should only allow DQfD to

see the reward in $1\%$ of the training steps. This would result in very poor performance (results not reported here).

## H   Unsuccessful ideas

In addition to the experiments presented in the main paper, we were unsuccessful at getting improvements from a variety of other ideas:

1. Deep RL is very sensitive to the scale of the reward, and this should be alleviated with quantile distributional RL while improving performance (Dabney et al., 2017). However, we did not manage to stabilize the training process with either distributional RL (Bellemare et al., 2017) or quantile distributional RL.

2. Both reward model and policy need to learn vision from scratch and presumably share a lot of high-level representations. Previous work has shown that training the same high-level representations on multiple objectives can help performance (Jaderberg et al., 2017). However, weight sharing between policy and reward model as well as copying of weights from the policy to the reward model destabilized training in our experiments.

3. To improve the reward models sample efficiency, we used parts of a pretrained CIFAR convolutional network as well as randomly initialized convolutional network. While this provided slight improvements in sample-efficiency on a few games, the effect was not very pronounced.

4. Since every observation from the environment is an unlabeled data point for the reward model, we could leverage techniques from semi-supervised learning to improve reward model's sample complexity. We tried applying the state of the art technique by Tarvainen and Valpola (2017) without much improvement in sample complexity while facing more training stability issues. Unfortunately it is unclear whether that particular approach does not work very well on the Atari visuals or whether the problem structure of reward learning is not very amenable to semi-supervision (for example because the reward is not very continuous in the visual features).

5. The bias toward expert demonstrations can limit the agent performance in environments where the expert performs poorly. We ran experiments where the large-margin supervised loss $J_E$ was gradually phased out during training. This had the desired effect of boosting performance in Enduro, where demonstrations are detrimental, but in games where demonstrations are critical, like Montezuma's Revenge and Private Eye, performance dropped along with the phasing-out of the supervised loss.

6. When using the expert demonstrations to augment preference annotation, we tried requesting annotations on pairs made up of two demo clips as a way to cover more of the state space in the reward model. It did not change the performance.

7. We also accidentally noticed that DQfD is extremely sensitive to small differences between the demonstration observations and the agent's experience—a small misalignment in the score-blanking method between demos and policy frames reduced the agent's scores to zero in the demo-heavy games. We attempted to increase robustness by adding noise and more regularization to the agent, but all such attempts hurt performance significantly.