[Reviews · NeurIPS 2018]

Reviewer 1



The paper presents a learning scheme for deep policies which is based on sequentially combining existing methods of learning from demonstration and learning from human preferences. The experiments validate the approach with simulated and real human teachers, obtaining interesting results but still remaining some open questions. It is said "Expert demonstrations are always kept in the buffer", so in cases wherein the demonstrations are not good, but the preferences of the teacher can lead to a high performance policy, there would be a problem, because the Demos are pulling the policy to a wrong area in the solution space. How to deal with this problem? Is it possible to draw from the experiments, in which kind of problems the preferences have a higher impact for the policy improvement. It would be interesting to see comments in a generalized way, about what kind of problems are more problematic for learning with human preferences, and which are prone to be easily trained (e.g. the kind of problems that can get superhuman performance). Results in Fig 3 (right) show something that can be really problematic in this presented approach. It is shown that the learned reward model is not aiming for the same objectives of the real reward function of the environment. So The RL agent always will try to maximize the reward function, however if it is wrong, obtaining high returns might be senseless (the method is not learning what the human teacher wants to teach, or the human understanding of the task is not complete). In those learning curves, the real return obtained at the end of the process is still an acceptable performance? It would be necessary more Why in these experiments the learned model r only takes as input the observations? is it for simplifying the model? or because it is known that the real score might be based only in states and not in actions? what would happen when considering the actions along with the observations in these experiments?

Reviewer 2



As the title implies, this paper examines imitation learning that combines human demonstrations and human preferences. The main algorithm builds on DQFD to learn Q-Values from human demonstrations and subsequently fine-tunes the policy using preference elicitation methods. More specifically, preferences are compiled into a surrogate reward function which is then used to further optimize the policy. The resulting algorithm is validated on nine Atari environments and results show that the technique of combining demonstrations with preferences is better than either using either source of feedback alone. Overall, the paper is clearly written, tackles a well-scoped problem, and presents compelling results. It's great to consider learning from more than a single type of human feedback and the combination of human demonstrations with preferences is a promising one. My largest concern is how reliant the algorithm is on preference feedback induced from the synthetic oracle. As the authors acknowledge, human preference feedback seems to nearly always perform worse than oracle preferences. It was unclear whether the problem stemmed from inexperienced human demonstrators or the simple fact that humans are bad at providing preference feedback. The latter makes me concerned about the ultimate usefulness of this approach. While it makes sense that the oracle preference feedback was used to automate the experimentation and remove the need for humans, it really detracts from the argument of learning on environments with no reward. I'm afraid the oracle could simply be helping the agent induce a learned version of the environment rewards. If this is the case, the proper point of comparison should be DQFD with access to true environment rewards. Another experiment that I would like to see would be a comparison between having human annotators spend time providing extra demonstrations versus preference labels. I wonder if the eventual algorithm performance would be better by directing humans to provide additional demonstrations rather than preferences. After reading the author rebuttal, my evaluation still remains at weak accept.

Reviewer 3



This paper combines a number of existing methods and performs an in-depth empirical evaluation. This is a very well written paper with high-quality scholarship. ---- I take issue with the ALE being "difficult enough to require nonlinear function approximation," i.e., "State of the Art Control of Atari Games Using Shallow Reinforcement Learning" https://arxiv.org/abs/1512.01563. I also think that the claim that it is "simple enough to run many of [sic] experiments on" is questionable, unless you're a large industrial research lab. The citations tend to heavily favor recent work, which is generally OK. But in the related work section, the citations imply that intrinsic RL was only invented in 2015 - it's much older than this. "clips" are discussed in Algorithm 1 on page 3 before being defined. It's not clear to me why expert transitions are always kept in the replay buffer (page 4) since the expert's demonstrations could be sub-optimal. In the majority of experiments a "synthetic oracle" that knows the true reward is going to select between clips. It's not clear to me whether this is a reasonable approximation of a human. Are there many clips where the reward is 23 vs. 24, or 2000 vs. 2003? Or are the clips normally qualitatively very different, and therefore easy for a human to compare without carefully estimating the reward on every action? --------------- After reading the authors' rebuttal and the other reviews, I have increased my score. Even though the assumption of an oracle (instead of a human) is strong, the work is still interesting. Additionally, I decided to weight the high quality of the paper's scholarship and writing more highly in my overall evaluation.